# Emerging responses implemented to prevent and respond to violence against women and children in WHO European member states during the COVID-19 pandemic: a scoping review of online media reports

Isabelle Pearson [1], Nadia Butler,[2] Zhamin Yelgezekova,[3] Åsa Nihlén,[3] Isabel Yordi Aguirre,[3] Zara Quigg,[2] Heidi Stöckl [1]

¹Global Health and Development, London School of Hygiene & Tropical Medicine, London, UK
²Public Health Institute, Liverpool John Moores University, Liverpool, UK
³World Health Organization Regional Office for Europe, Copenhagen, Denmark

**Correspondence to**
Isabelle Pearson;
isabelle.pearson@lshtm.ac.uk

## ABSTRACT

**Objectives** This study aims to explore the strategies that governments and civil society organisations implemented to prevent and respond to the anticipated rise in violence against women and/or children (VAWC) during the 2019 novel coronavirus (COVID-19) pandemic.

**Design** A scoping review and content analysis of online media reports.

**Setting** WHO European region.

**Methods** A scoping review of media reports and publications and a search of other grey literature (published from 1 January to 17 September 2020). Primary and secondary outcome measures included measures implemented by governments, public services and non-governmental and civil organisations to prevent or respond to VAWC during the early months of the COVID-19 pandemic.

**Results** Our study found that in 52 of the 53 member states there was at least one measure undertaken to prevent or respond to VAWC during the pandemic. Government-led or government-sponsored measures were the most common, reported in 50 member states. Non-governmental and other civil society-led prevention and response measures were reported in 40 member states. The most common measure was the use of media and social media to raise awareness of VAWC and to provide VAWC services through online platforms, followed by measures taken to expand and/or maintain helpline services for those exposed to violence.

**Conclusion** The potential increase in VAWC during COVID-19-imposed restrictions and lockdowns resulted in adaptations and/or increases in prevention and response strategies in nearly all member states. The strength of existing public health systems influenced the requirement and choice of strategies and highlights the need for sustaining and improving violence prevention and response services. Innovative strategies employed in several member states may offer opportunities for countries to strengthen prevention and responses in the near future and during similar emergencies.

## Strengths and limitations of this study

▶ This is the first study to use a systematic methodology to conduct a media search on measures taken to prevent and respond to violence against women and children during COVID-19.
▶ This study was able to extensively search media in the Russian language and therefore provide broad coverage of many countries in Central Asia and Eastern Europe.
▶ Use of only English and Russian search terms means that media reports from countries where these languages are widely spoken will be over-represented in the results.
▶ Focusing largely on media reports means that innovative and unprecedented measures will be more frequently reported, thus countries which had strong baseline measures in place already may be under-represented.

## INTRODUCTION

Violence against women and children (VAWC) is an important public health, gender equality and human rights issue.[1] Based on estimates by the WHO, around 25% of ever partnered women in Europe have experienced sexual and/or physical violence by a partner[2] and 12% of children aged 2–17 years were reported to have experienced violence in the past 12 months (around 15.2 million children).[3 4] During a pandemic, VAWC is likely to increase due to stress related to economic insecurity, quarantine and social isolation, disruptions in health and social services and increased house and care work while working from home.[5 6] The outbreak of the 2019 novel coronavirus (COVID-19) has been no exception. As more and more



countries went into lockdown or imposed other physical distancing restrictions, there were increasing anecdotal reports from countries of a rise in VAWC.[6 7]

Alongside other UN agencies, WHO has issued clinical and policy guidelines on responding to violence against women (VAW) and violence against children (VAC), including during COVID-19.[8] For governments and policy-makers, they recommend that violence prevention and response is considered in all response plans and mitigation methods, information dissemination to the public regarding available services and increased risk factors, as well as to alert essential service providers within the community to make them aware of signs that indicate violence. They recommend governments enforce rules and regulations around key risk factors for violence such as alcohol, drugs and weapons and to work across sectors and with civil society to coordinate support, including referral services. To support survivors, WHO recommends governments make provisions to allow those seeking help to safely leave the home, ensure and expand helpline functions and identify ways of making services accessible remotely. The WHO guidance also includes advice to health system managers and health providers on how to focus on VAWC in the organisation and delivery of health services, in data collection and through preventive health measures. Recommendations include providing information about services available locally that continue to offer first line support and medical treatment for survivors of violence through the first points of contact in health facilities, in line with WHO recommendations.[8]

With these WHO recommendations on preventing and responding to VAWC during COVID-19 as a basis, the aim of this study was to investigate what measures were taken by governments, Non-governmental organisations (NGOs) and other civil society organisations across WHO European member states in the context of the COVID-19 pandemic, during the first 9 months of 2020.

## METHODS

This study is based on a scoping review of media reports, journal articles and official organisational reports, including information from webinars and other expert meetings. Searches were conducted in English and Russian. For any article, study or report to be included, they had to include data on at least one measure implemented in the context of the COVID-19 pandemic, either in response to or to prevent VAW and/or VAC. Reports also had to have been published between 1 January 2020 and 17 September 2020 and provide data for any member state of the WHO European Region, including the self-governing territories of Kosovo and Greenland, or data that encompassed multiple countries including at least one of the aforementioned countries. The search excluded reports that did not provide at least one measure or only recommended measures without stating that they had been implemented.

### Media review

A review of media reports published between 1 January 2020 and 17 September 2020 was conducted. A search of online news reports in English was conducted using the International Newsstream of ProQuest, limited to the Asian, European and Middle Eastern news streams. The search strategy included terms related to COVID-19 and VAWC and was restricted to the WHO European member states, with no language restrictions. A full search strategy can be found in the online supplemental methods 1. Reports were all exported into Excel and screened at the full-text stage.

For the Russian language media reports, searches were conducted using East View Information Service. To increase coverage, an additional search was conducted in Yandex.ru. A full search strategy for the Russian-language reports can be found in online supplemental methods 2. A hand search was conducted for Radio Liberty's Central Asian branches, as well as Sputnik News. Results of the hand searches were exported manually, the reviewer screened the abstract/title and exported those deemed relevant into the Excel sheet. Those included at the title and abstract stage were then screened in further detail for inclusion at the full-text stage.

### Publication review

A search was conducted to review any publications that were published between 1 January 2020 and 17 September 2020 concerning the COVID-19 pandemic and VAWC. For the non-Russian language articles, this search was conducted in OVID, restricted to Ovid MEDLINE(R) and In-Process & Other Non-Indexed Citations and Daily and using the search strategy outlined in online supplemental methods 3, with no language restrictions. Publications from OVID were imported into the Excel sheet to be reviewed at the title/abstract stage and if relevant, screened at the full-text stage. A separate search was conducted in Russian using Google Scholar with a translation of the search strategy. Google Scholar was linked to eLibrary.ru and CyberLeninka. Full texts that met the inclusion criteria were extracted into the Excel workbook. High-level organisational websites were searched in both English and Russian and any reports that met the inclusion criteria were also included for the full-text review, along with any relevant information collected from organisational webinars that occurred during this time and were attended/accessible to the research team.

### Quality assurance

Three reviewers contributed to the screening of the media reports and publications and data extraction. A trial screen and data extraction were conducted where each reviewer screened and extracted data in parallel for a subset of search results. The results of the trial screen were compared between reviewers and, once consistency was confirmed, each reviewer separately screened and extracted data from an allocated percentage of the database search results. Overall, 10% of the data extracted by

each reviewer, excluding the Russian language search, were checked by a second reviewer to ensure consistency.

## Data extraction and management

All included results were stored and analysed within an Excel workbook. For each measure, the description of the strategy was extracted and subsequently each strategy was labelled 'government led and/or sponsored' or 'NGO and civil society'. Where a measure was led by a government and at least one NGO or civil organisation, the measure was labelled as 'multiple' (please note: measures led by local NGO and civil organisations along with external country governments were labelled as 'NGO and civil society', United Nations (UN) agencies were considered as 'NGO and civil society' and European Union (EU) funded measures were considered 'government led and/or sponsored'). Where the measure's leading body was unclear from the extracted text, a Google search was conducted for clarification, and if it remained unclear then it was labelled 'unclear'. Each unique measure contained in the text extract was also assigned a topic code. Topic codes were created through thematic analysis at the data extraction stage as the researchers' understanding of the types of strategies used were developed. These topic codes were used to group the data with similar responses. Where responses fell under multiple topic codes, the reference was duplicated so that it could fall under each code. The country, region, date of publication and the source of the data (where available) were also extracted.

## Patient public involvement

No patient or public member was involved in the study.

## RESULTS

The ProQuest search for media reports yielded 1610 reports (after removing duplicates) to be included in the full-text search. The Russian-language media search yielded 95 results that were considered for full-text screening. Of the 1705 screened reports, 187 media reports were included in the analysis (exclusion reasons are displayed in online supplemental table 1). The publication search using the OVID database yielded a total of 157 publications, out of which 6 were included in the analysis. The Russian language database search resulted in seven publications, of which one was included. The hand searching of both Russian and non-Russian language grey literature reports of international organisations and NGOs and other civil society groups resulted in 158 additional unique reports to be included in the study. The majority of measures reported in this study are from media reports and we rely only on the information those media reports provided. The research team did not conduct further checks on the information provided. The publications found did not include any primary data collection regarding measures to prevent and respond to VAWC during COVID-19. They did, however, cite

additional measures announced through media platforms, most often online newspaper articles. Due to this, the publications were used to supplement the results of our media search with the additional media reports cited through the publications and did not offer any examples of primary data analyses.

## Responses implemented to protect women and children from violence during the COVID-19 pandemic

Our study found reports to illustrate that in 52 out of 53 WHO European member states, at least one measure to prevent or respond to VAWC had been implemented in the context of the pandemic. The most common types of measures were government-led or government-sponsored measures, of which at least one example was reported for 50 of the 53 member states and in territories Kosovo and Greenland. NGO and civil society-led responses were reported at least once in 40 of the 53 member states and in the territory of Kosovo. table 1 displays the full breakdown of the types of measures across countries, with further details reported in online supplemental table 2.

Most media reports used the term domestic violence, often without clarity as to whether the term was used to encompass both children and adults. Unless stated otherwise, in this report, the term domestic violence is interpreted to cover both women and children. Out of the 53 member states and two additional territories included in the study, all but one country reported measures to prevent or respond to 'domestic violence', 33 (60%) implemented measures to prevent or respond to VAW explicitly and 32 (58%) to VAC explicitly.

## Government-led or sponsored measures

The most frequently reported national or regional government measures were media and other information dissemination campaigns, reported in 39 member states through various media channels. The Irish Department of Justice, for example, collaborated with frontline services to develop television, radio and social media advertisements highlighting their continued support for those subjected to abuse.[9] In Slovenia, the city of Ljubljana distributed leaflets with VAWC NGO contact numbers, along with the digital broadcasting of violence hotline numbers on public screens.[10]

Regarding high-level intersectoral approaches, governments at the national and regional level created government task forces to help prevent and respond to VAWC during COVID-19. In Belgium, the Brussels and Walloon governments created a task force in order to monitor shelters, assist actors in the field, rapidly identify emerging needs and spread information and awareness.[11] The government of Luxembourg created a crisis management system to mitigate the risk of increased VAWC.[12]

A number of countries dedicated specific funding for measures to prevent and respond to VAWC, including the UK, France, Israel, Ireland, Italy, Sweden and Kosovo.[11 13–18] The government of Iceland also included

**Table 1** Reported measures in WHO European member states, including territories Kosovo and Greenland (categorised by type, number of countries and percentage of countries)

| Implemented measure | Number of countries in which measure was reported, overall (%) | Number of countries in which measure(s) were reported as led/sponsored by government (national) | Number of countries in which measure(s) were reported as led/sponsored by government (regional) | Number of countries in which measure(s) were reported as led by NGO/civil society | Number of countries in which measure(s) were reported as led by NGO/civil society and government in partnership | Number of countries where leading body was unclear |
|---|---|---|---|---|---|---|
| **Response services** | | | | | | |
| Helpline expansion/maintain | 43 (78.2) | 33 | 3 | 20 | 2 | 17 |
| Shelter expansion/maintain | 38 (69.1) | 25 | 8 | 14 | 6 | 12 |
| Move resources online | 38 (69.1) | 15 | 2 | 25 | 4 | 10 |
| VAWC app | 13 (23.6) | 11 | 0 | 5 | 0 | 2 |
| Financial/goods support | 13 (23.6) | 5 | 1 | 9 | 1 | 1 |
| Extra funding for NGOs | 13 (23.6) | 11 | 1 | 3 | 0 | 0 |
| Proactive contact with survivors | 11 (20.0) | 8 | 2 | 3 | 0 | 0 |
| Emergency mobile teams | 3 (5.5) | 1 | 0 | 1 | 1 | 0 |
| Monitor past perpetrators | 3 (5.5) | 0 | 0 | 3 | 0 | 1 |
| EU funding earmarked for VAWC services | 2 (3.6) | 2 | 0 | 0 | 0 | 0 |
| Testing for survivors | 1 (1.8) | 1 | 0 | 0 | 0 | 0 |
| Open air F2F appointments | 1 (1.8) | 0 | 0 | 1 | 0 | 0 |
| **Awareness and outreach** | | | | | | |
| Media campaign/Info dissemination | 48 (87.3) | 37 | 6 | 25 | 4 | 8 |
| Official guidance/policy | 30 (54.5) | 26 | 0 | 0 | 9 | 0 |
| Advocacy | 11 (20.0) | 3 | 2 | 8 | 0 | 0 |
| Creation of taskforce | 7 (12.7) | 6 | 1 | 0 | 1 | 0 |
| Community/corporate Fundraising for VAWC services | 3 (5.5) | 0 | 0 | 3 | 0 | 0 |
| **Legal support for survivors** | | | | | | |
| Fast track/prioritise/extend legal processes | 15 (27.3) | 13 | 0 | 2 | 1 | 3 |

**Table 1** Continued

| Implemented measure | Number of countries in which measure was reported, overall (%) | Number of countries in which measure(s) were reported as led/ sponsored by government (national) | Number of countries in which measure(s) were reported as led/ sponsored by government (regional) | Number of countries in which measure(s) were reported as led by NGO/ civil society | Number of countries in which measure(s) were reported as led by NGO/ civil society and government in partnership | Number of countries where leading body was unclear |
|---|---|---|---|---|---|---|
| Exempt from lockdown measures* | 13 (23.6) | 11 | 1 | 1 | 0 | 2 |
| Eviction of perpetrator | 5 (9.1) | 3 | 0 | 0 | 0 | 3 |
| Police prioritise DV cases | 3 (5.5) | 3 | 0 | 0 | 0 | 0 |
| Alcohol ban | 1 (1.8) | 0 | 1 | 0 | 0 | 0 |
| Coordination with other services | | | | | | |
| Pharmacy help point | 14 (25.5) | 3 | 3 | 6 | 1 | 8 |
| Raise police awareness | 9 (16.4) | 8 | 0 | 1 | 1 | 0 |
| Supermarket/ shop help point | 5 (9.1) | 2 | 0 | 3 | 0 | 2 |
| Ensure provision of existing SRH services | 5 (9.1) | 2 | 1 | 0 | 0 | 2 |
| Childcare provision | 4 (7.3) | 1 | 1 | 1 | 0 | 1 |
| Postman check in | 3 (5.5) | 3 | 1 | 1 | 0 | 1 |
| Free transport | 2 (3.6) | 0 | 0 | 2 | 0 | 0 |
| Police codeword | 1 (1.8) | 1 | 0 | 0 | 0 | 0 |
| PPE for police | 1 (1.8) | 1 | 0 | 0 | 0 | 0 |
| Dentist guidelines for telephone assessment | 1 (1.8) | 0 | 0 | 1 | 0 | 0 |
| Medical care for refugees | 1 (1.8) | 1 | 0 | 0 | 0 | 0 |
| Other service coordination | 3 (5.5) | 0 | 0 | 2 | 2 | 0 |
| Strengthen capacity/protection for professionals | | | | | | |
| PPE/testing for VAWC staff | 11 (20.0) | 7 | 0 | 5 | 0 | 1 |
| Other support for VAWC centre staff | 3 (5.5) | 1 | 0 | 2 | 1 | 0 |

*'Exempt from lockdown measures' refers to a situation where those facing violence within the home were exempt from the strict lockdown measures imposed in their local area, for example, being exempt from curfews or being allowed to use public transport to access support services.

DV, Domestic Violence; F2F, Face-to-face; NGO, Non-governmental organisations; PPE, Personal Protective Equipment; SRH, Sexual and Reproductive Health; VAWC, violence against women and/or children.

investment in efforts to combat domestic violence in their national financial aid package announcement.[19]

Various government guidance packages and policies were announced, either specifically for VAWC or with measures to respond to VAWC included, across member states (see online supplemental table 1 for details). This included the creation of a committee in Israel to examine the national incidence of women killed during lockdown,and an interagency communication strategy on VAWC during the COVID-19 crisis in Georgia.[11 20]

In terms of service coordination, multiple examples were found of government-led measures for VAWC, primarily supporting the maintenance and expansion of VAWC helplines and shelters. Methods varied from the introduction of new helplines numbers, such as text messaging numbers introduced in France and Israel, to the introduction of WhatsApp services in Spain.[11 14 21] The maintenance of shelters was also reported as a key priority for governments during this time, with measures taken to ensure they remained open or were expanded, for example, by providing them with hotels or additional accommodation.[22–28] A few governments also provided personal protective equipment (PPE) and COVID-19 testing for staff and survivors and declared shelter staff as essential workers to exempt them from lockdown measures.[11 29–32]

In terms of moving VAWC resources such as psychosocial support and counselling online, 15 governments (14 member states plus Kosovo) announced technology-based solutions, including a new government-led email address accessible to survivors and professionals in Portugal and the French online resource, stopblues.fr, that provides support and aids reporting of violence.[11 33]

Pharmacies and supermarkets were also used to spread information; some governments sought to physically disseminate information such as numbers for VAWC helplines and relevant service providers, while other governments implemented pop-up counselling centres in some supermarkets.[11 24 34] Pharmacies, specifically, were also encouraged by governments to participate in the European scheme using the 'Mask-19' code word, whereby pharmacy visitors could mention 'Mask-19' if they required help for domestic violence.[11 35–38]

Several member states reported working with the police to address the expected rise in VAWC during COVID-19. In Ireland, the police service proactively contacted every previous survivor of domestic abuse known to them.[39] The government of Andorra created a video tutorial updating police officers on the VAWC guidelines protocol[11] and the Norwegian police implemented a comprehensive set of measures to react to changes in the levels of violence and ensure that police would focus on VAWC cases.[11] To ensure survivors could access required services, apps were identified for them to use to contact the police; in Czechia, the government disseminated the 'Bright Sky' app to allow survivors to contact support organisations and the police and also to access advice and store evidence.[11]

Other measures focused on improving legal processes and provisions include the fast tracking, prioritising and extension of legal processes surrounding cases of VAWC. In particular, in Croatia, allowances were made for court deadlines missed by survivors due to COVID-19; while in Serbia, the High Court Council declared that despite the courts closing, domestic violence cases would continue to be processed.[11 40] In the Russian Federation, newspapers reported the introduction of mandatory reporting, obliging the police to investigate cases of VAWC even without an official request from the survivor.[41] In several other countries, governments introduced rules to ensure that in situations of abuse the perpetrator is evicted instead of the survivor.[11 42–44] In Kyrgyzstan, a bill was passed increasing detention of perpetrators of domestic violence from 3 to 48 hours.[45] Policies around releasing prisoners early or granting pardons under the pandemic situation excluded prisoners convicted on VAWC charges from release in several member states.[46–51]

Some government measures were targeted at improving the response to those experiencing violence. In many countries facing stricter lockdown rules, such as Italy, Spain and Kosovo, governments announced that the stay home orders and strict curfews did not apply to survivors who were seeking support.[42 43 52] Furthermore, in Malta and Ireland, women and children known to be trapped in abusive homes were reportedly provided rent supplements.[53 54] Governments in Czechia, France and the UK also promoted the use of couriers and postmen to check in with survivors of abuse.[11 18 27]

In some cases, governments and NGOs jointly led measures to prevent and respond to VAWC. A technical group was formed in Montenegro within the European Union and UN Women-led regional programme to prepare guidance for institutions on how to deal with VAWC during COVID-19.[11] In Albania, the Women Forum Elbasan collaborated with state police to allow beneficiaries to be accompanied by police patrol, ensuring that they could access VAWC services despite the lack of transport available during the pandemic.[55] Furthermore, in Uzbekistan, a joint project with the United Nations Development Programme and the Ministry for Supporting Mahallas and Families prepared flyers that were distributed to pharmacies in Tashkent City and Tashkent region of Uzbekistan in order to reach vulnerable groups of the population and provide them with referral numbers in case of violence.[56] Measures that were not attributable to an organising body but picked up in the search included the printing of VAWC helpline numbers on milk bottles in Germany.[57] In Sweden, a popular landlord company distributed flyers with VAWC-related information to all its tenants.[57]

## NGO-led and/or civil society-led measures

Media campaigns and information dissemination were also one of the most frequently reported measures used by NGOs and civil organisations to prevent and respond to VAWC during COVID-19, reported across 24 member

states plus Kosovo. Their strategies varied but were most often based around radio, television and social media. For example, the Union of Women Associations of Heraklion in Greece ensured a constant presence on TV shows, news channels and radio commercials to spread awareness of VAW during COVID-19.[38] An online awareness campaign, Stopfisha, was launched via social media in France as a response to the suspected rise in revenge porn as a result of lockdown. This campaign helped to find survivors of abuse and assist them in reporting it.[58 59] As well as raising awareness and providing service contact details, social media was also used by NGOs and civil organisations to help those facing VAWC to seek help. For example, in Poland, a fictitious online cosmetic store was set up through Facebook where survivors of domestic violence could request help by pretending to order goods.[60]

Similar to the use of social media, NGOs and civil organisations in 24 member states plus Kosovo used online methods, and five member states developed apps, in order to facilitate access to VAWC support services and/or to provide them with psychological and legal support through online platforms such as Zoom and Skype. NGOs across member states provided services via Telegram, WhatsApp and Viber.[18 61–66] The NGO SPAVO in Cyprus purchased 35 smart watches for women facing domestic violence that had built-in safety alarms.[67]

The maintenance and expansion of helplines and helpline services were key measures taken by NGOs and civil society groups across 19 member states, plus Kosovo. NGO-led shelters in Ireland expanded capacity by collaborating with Airbnb, in France a sports stadium was used and in Italy collaborations emerged with Booking.com and also a former convent.[23 68 69] In the Republic of Moldova, to overcome government-imposed quarantine measures that meant shelters were unable to accept new residents, the NGO Promo-LEX rented an apartment.[13] To ensure that helplines remained functional and could meet demand, measures included increasing the number of helpline volunteers, creating chat/SMS messaging options and expanding helpline hours.[11 55 70–75] New helplines were also set up by NGO and civil organisations in several members states.[11 74 76–80] In France, the new helpline, 'Don't Hit', was launched in April to provide counselling and specialist psychological assistance to perpetrators of violence.[11] Furthermore, systems were reportedly set up to allow helpline staff to work from home and PPE and/or COVID-19 testing were made available to VAWC centre staff in several countries.[11 55] The NGO 'CAM Firenze' in Italy organised fortnightly peer-support meetings to help staff with their emotional management and well-being during the pandemic.[72] In addition, one Belgian NGO set up open air face-to-face appointments, where survivors could receive support while on a socially distant walk with centre staff.[38]

Similar to the previous measures taken by governments to use pharmacies and supermarkets as help points for survivors, NGOs in North Macedonia, Italy and Greece also used them for the physical distribution of leaflets and posters with relevant VAWC service provision details.[38 55 81]

Other NGO and civil organisation measures conducted in only a few countries include proactive contacting and monitoring of both perpetrators (Luxembourg and Slovenia) and known survivors (Serbia, Republic of Moldova, Italy and Belgium).[11 38 55 72] Furthermore, food and/or hygiene packages were distributed to vulnerable populations, including those affected by violence, in Malta, Naples (Italy), North Macedonia and Albania and financial and/or in-kind support was provided to vulnerable groups in Bosnia and Herzegovina, Spain and Montenegro.[11 55 82–85] Other forms of support include the provision of free rail travel by a rail company in the UK (in partnership with the NGO Women's Aid) and in Serbia, where public transport was abolished during the pandemic, multiple NGOs provided private transport for survivors to access their facilities.[55] Furthermore, UK-based dentists were called on to update their guidelines for telephone assessments under lockdown to help them continue to assess the risk of VAWC in patients with facial injuries.[86]

Measures to prevent and respond to VAWC were also implemented by international organisations. In Kosovo, body cameras were provided by UN Women to the police's domestic violence department to ensure sufficient evidence was collected during call outs.[87] In Serbia, the United Nations Development Programme supported public prosecutors from eight prosecution districts to organise online multiagency meetings to process cases of VAWC.[40] The All-Ukrainian Charity Foundation, UNICEF and the United Nations Foundation supported sociopsychological assistance mobile crews in the Ukraine.[88]

## Measures specifically to prevent VAC

The implemented measures described above were often summarised under the headings of domestic violence, yet were more focused on addressing VAW rather than VAC. Only a few measures were identified that addressed VAC specifically, for example, the government of France set up services to allow children facing situations of violence to directly contact authorities via SMS.[89] In Ukraine, a chatbot was set up via Telegram to provide answers to common GBV-related questions and facilitate communication with state legal aid workers, this measure was reportedly popular among teen users.[90] In Sweden, the social media app 'Snapchat' was used by service providers to raise awareness and reach out to those aged 13–21 years who were potentially at risk of abuse. Further, in France, a children's aid charity partnered with gaming platform Fortnite to allow NGO volunteers to roam the virtual battle arenas; players could reach out to them privately to report abuse and ask for advice and assistance.[91] Those services all targeted older children and teenagers and no reports could be found identifying measures addressing and preventing VAC among younger children.

## DISCUSSION

This study provides an initial overview of some of the measures taken by the member states of the WHO European region to prevent and respond to VAWC during the COVID-19 pandemic. Government responses were identified for 50 out of the 53 member states of the WHO European region; NGOs and civil organisation-led measures were identified in 40 member states in the Region. The most frequently reported measures found in the study were media campaigns and other forms of information dissemination. Ensuring women and children are aware of the resources available to them and how the situation of the pandemic may affect their access to such resources is crucial and aligns with WHO recommendations. The high frequency of these measures may be due to political factors and the fact that they can be rapidly arranged and implemented with a relatively low financial or resource burden. A WHO baseline assessment on health system responses to VAW in the WHO European region, published in 2019, shows a very high proportion of WHO European member states have made national policy commitments to eliminate VAW and had national or subnational multisectoral action plans to prevent and respond to VAW. This lays the foundation for information dissemination and media campaigns on this topic. The same assessment further found that a specific budget line for financing a health system response to VAW only existed in five European countries.[92]

The majority of the results of this study pertain to *response* measures. Media campaigns and information dissemination were some of the only reported *prevention* measures and existing evidence suggests that they are rarely effective on their own.[93] This is especially pressing given the long-term impact of the pandemic on poverty and economic well-being, both factors that are known to be linked to VAWC.[94] As very few preventative measures were identified by this report, it suggests that they were not only rarely implemented, but also that existing prevention measures may have also been halted during the pandemic. Without the continuation of prevention methods, the pandemic will have far-reaching and long-term impacts on VAWC and due to this, the measures taken to respond to VAWC during this time should be continued long into the future to mitigate these long-term impacts.

The second most frequently reported group of measures taken by governments was service coordination for survivors which included the maintenance and expansion of shelters and helplines and online services for women and children facing violence. This is in line with the UN recommendations that shelters should be classified as essential services during COVID-19 and that women should have access to safe ways to seek support.[95] The need to expand or strengthen VAWC services during a pandemic could be dependent on how well these services were already supported prior to the pandemic, as well as the nature of the lockdown measures. Several countries pledged additional funds to civil society organisations to respond to the increased service needs, but our findings currently do not reveal if countries also allocated investments to strengthen public health systems service provision. Some countries took measures to facilitate physical access to services, while other moved services online or through apps.

Many of the measures taken by governments and NGO/civil organisations included the continuation of counselling and psychosocial support, predominantly through the move to online methods of communication, which follows the WHO recommendation of ensuring women-centred interventions for survivors.[93] However, measures ensuring that services are available and accessible online were more often the work of NGOs and civil society organisations than of governments. Besides the movement of resources online, our study found other examples where certain measures were more often reported as being NGO and civil organisation led than government led, for example, the provision of financial and good packages. Such provisions included money, food and hygiene-related items and were most often provided to groups identified by the NGOs as vulnerable. Although not well represented in the results of this study, vulnerable groups will have likely been some of the worst affected populations during the pandemic, not only regarding VAWC. The fact that support to vulnerable groups was most often reported as NGOs/civil organisation-led highlights how such populations are often under-represented in government-led responses and the important role civil society has in filling these gaps. Non-governmental/civil society organisations also often had to mitigate the negative impact of government-led pandemic responses. One example being the provision of transport in several member states where, in the absence of government allowances for certain groups to break lockdown rules, civil society organisations had to provide alternative means of transport. Furthermore, NGOs and civil organisations were also frequently reported as leading advocacy campaigns aimed at raising awareness to VAWC in the context of the pandemic and demanding action. This highlights civil society as a crucial force for focusing attention to human rights issues and holding governments accountable.

From the results of this study, it is clear that measures identified to prevent and respond to VAWC were predominantly focused on VAW. The study did not identify any school-based measures and there were very few parenting programmes for the prevention and response to VAC. School-based interventions and parenting programmes are known to be effective in the prevention and reduction of VAC;[93 96] however, with widespread school closures across Europe, school-based interventions were not feasible. Further, with the current additional burden of homeschooling placed on parents during COVID-19, the possibility of conducting parenting programmes during this time was low. Schools also provide a crucial pathway for the identification and response to VAC and therefore, with schools closed, it is likely that the majority of child abuse will have gone unreported during this time. As schools

across many of the member states have now reopened, this will be a crucial time to ensure that children are offered the necessary support and services that they may not have had access to under the pandemic restrictions. Furthermore, it is vital to consider the long-term impact that the pandemic will have had on children who have spent lockdown at home with abusive family members. VAC and VAW are both forms of domestic violence and share many overlapping risk factors, however, prevention and response measures for the two types of violence can differ greatly. Therefore, it is important that the need to address each separately is not overlooked when both are grouped under the term 'domestic violence'. Our analysis most often only identified measures to address VAWC among the general population, not considering those who are particularly vulnerable to violence, such as migrant women, sex workers or low-income families, who might also be affected more by COVID-19 restrictions.

To contextualise the results, there are some limitations of this study that should be considered. First, the attention of media reports is often biased towards new and innovative strategies rather than the actions taken to maintain existing responses and systems. Therefore, countries with strong existing frameworks for VAWC prevention, which did not need to rely on novel responses, are very likely to be under-represented in these data. Second, the use of only English or Russian search terms will have excluded reports in other European languages. Due to this, some countries will be over-represented in these results, particularly the UK, the Russian Federation and Ireland. This varied representation, combined with the focus on innovative responses, means that a lack of data reported for a country should not be interpreted as a country's lack of action to protect women and children from violence during COVID-19. Similarly, due to the lack of evaluation of the presented measures, the frequency in which measures are reported here is not an indication of their effectiveness. For example, Greenland was the only jurisdiction that our study found to have enforced a regional alcohol ban for the specific purpose of preventing VAC, but due to the wealth of evidence linking alcohol consumption to VAWC, it is likely that this measure would have had a positive effect on the reduction of violence.[97 98] Furthermore, the newspaper articles do not give any indication at this stage on whether the measures had the proposed positive effect, had any adverse side effects and what level of training and support was needed to implement them. For example, using postmen to check on women at risk of VAW would require substantial training in VAW, being non-judgemental, ethics and safety. Finally, the search strategy only focused on reports explicitly referring to VAWC, thereby ignoring upstream preventative measures such as parental support, education and childcare provision that are not always mentioned in the context of VAC. This may explain the under-representation of child-focused responses in our results. Similarly, focusing only on reports that referred to violence means that many service-based measures were not widely represented in

our results, such as measures to ensure access to sexual and reproductive health services, continued abortion care and access to HIV care and prophylaxis for sexually-transmitted infections, all key aspects of clinical care when responding to VAW.[99]

## CONCLUSION

These results provide evidence that a diverse set of measures were taken by European governments, NGOs and civil society organisations to maintain and expand VAWC service provision during the first months of the pandemic. While it is clear that the COVID-19 context has led to an increased focus on VAWC and that an overwhelming majority of countries have taken some kind of action, further research is needed on the impact of these actions and what can be learnt from the past couple of months in order to improve future responses. For some women and children, the living situation during lockdown is, unfortunately, not too dissimilar to pre-pandemic life living with an abuser. Therefore, rather than suggesting the implementation of the outlined emergency measures alone, governments should be encouraged to reflect on the gaps in existing national VAWC response frameworks in their countries, in particular, ensuring greater emphasis on VAC within measures to prevent and respond to VAWC. So, while we should celebrate the ability of governments, NGOs and civil society organisations to rapidly adapt under pressure, the responsibility should now be on governments to develop stronger baseline support systems to ensure that the responsibility to protect women and children does not fall on NGOs and civil society as the world rebuilds from COVID-19. Further research is also needed to understand how the mental health impacts of the pandemic along with the wider disruptions to service provision and access to work and education will affect VAWC in the long term.

**Contributors** All authors contributed to the development of the study protocol and the planning of the paper, provided feedback on all drafts of the paper and edited the final manuscript. IP, NB and ZY conducted the media, publication and grey literature searches (IP and NB in English, ZY in Russian) and content analysis. IP was the lead author of the article. All authors approved the final manuscript.

**Funding** This work was supported by WHO Europe grant number WHO Ref 2020/1021489.

**Competing interests** None declared.

**Patient consent for publication** Not required.

**Provenance and peer review** Not commissioned; externally peer reviewed.

**Data availability statement** All data relevant to the study are included in the article or uploaded as supplementary information.

**Author note** All references to Kosovo in this document should be understood to be in the context of the United Nations Security Council resolution 1244 (1999).

**ORCID iDs**
Isabelle Pearson http://orcid.org/0000-0001-8857-5703
Heidi Stöckl http://orcid.org/0000-0002-0907-8483

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
