## [Reviewer comments · BMJ Open]

ARTICLE DETAILS

TITLE (PROVISIONAL)	Emerging responses implemented to prevent and respond to violence against women and children in WHO European Member States during the COVID-19 pandemic: a scoping review of online media reports
AUTHORS	Pearson, Isabelle; Butler, Nadia; Yelgezekova, Zhamin; Nihlén, Åsa; Yordi Aguirre, Isabel; Quigg, Zara; Stöckl, Heidi

VERSION 1 – REVIEW

REVIEWER	Rachel Jewkes South Africa Medical Research Council, South Africa
REVIEW RETURNED	29-Oct-2020

GENERAL COMMENTS	Thank you for asking me to review this paper. It reflects an interesting project that has been undertaken amazingly quickly. I have two more substantial comments. The first relates to the framing as VAWC. As far as I can tell from the text there were very few measures that specifically were designed to reach out to children and where these existed they seem to be for rather older children. There is some discussion of lack of work identifying abused children in school as these were closed and on parenting, however this seems rather uncomfortably placed in the paper which clearly doesn't refer much to measures for children at all. The question for me is: was very little done about VAC or was it missed in the search? Perhaps not so often reported in papers? My perception is that VAC has been little discussed in the context of lockdown and these findings seem to support that. I think this is rather an important problem (for children) and that the authors should find a way of separating discussion of measures for VAW and VAC in the results so that the lack of VAC measures becomes more visible. Further just how much of a problem this will be for children (locked at home with abusers) should be separately discussed and recommendations framed. The second comment relates to VAW prevention. Most of the measures were ones that generally are considered 'response measures'. They may have enabled some individual women experiencing violence to get help, but other prevention programming this year has been largely halted across the world and none of these measures were directed at getting men to stop being abusive. Of great concern is that poverty is a potent driver of gender inequality and violence and this has increased this year. The main 'prevention' measure was awareness campaigns, which on their own have never been shown to prevent VAWC. I feel these concerns need to be built into the paper, as the suspension of work on prevention of VAWC, in the context of loss of gains on risk factors, will have a long term impact on the problem globally. Table
--

	Table 1 is first referred to in the methods and since it is conventional to place a Table after its first reference this is unsatisfactory. I suggest revising the section to remove reference to the table there so it can be clearly displayed in the results. In Table 1: I am not clear how 'EU funding' is a measure – what was the EU funding for? Similarly I am not sure that community/corporate fundraising is a measure – again the question is – for what? Minor edits The paper has a number of typographical errors, a few have been identified below. It would benefit from a thorough edit. Delete the second 'measures' in line 16 abstract Article summary line 5 'a' before systematic Line 13 – change 'meaning' to 'thus' Page 5 line 3 'VAWC' is an important...issue. Line 7 change 'or' to 'that is' Line 9 'quarantine' Line 23 delete 'to' after governments before enforce Line 25 delete 'to' after governments before 'make' Line 32 change 'to provide' to 'providing' Page 6 line 2 change 'studies' to 'reports' Page 8 line 1 'of' Kosovo Page 8 sentence covering lines 6-11 needs editing Line 17 – is it Justice and Equality? Line 31 delete 'have' Line 36 'were' rather than 'have been' Line 38 delete 'who have being' Page 9 line 2 'technology' rather than 'technical' Line 7 is 'Telegram' correct here? Line 12 – is it numbers of relevant service providers? Line 15 'using' rather than 'of' Line 20 full stop after 'them' new sentence for the Andora example Line 32 – delete 'the' and edit 'legal processes' Line 40. Full stop after 'survivor' delete 'and' and capital for 'In' Line 44 delete 'a' Page 10 line 13 add 'the' before 'social media' – who was it who checked in? was it really 'the app'? if so how did that work Page 12 line 12 insert 'often' before 'reported' Line 23 'service provision' Line 41 insert 'was' before 'most often' Page 13 line 3-7 'As schools across the member states are now reopening, this will be a crucial time to ensure that children are offered the necessary support and services that they may not have had access to under the pandemic restrictions.' This sentence needs editing to take into account the fact the paper is being considered for publication in a different point in time
--	---

REVIEWER	Dr. Jimmy Ben Forry Kampala International university Uganda
REVIEW RETURNED	03-Nov-2020

GENERAL COMMENTS	The work contained herein is scientifically sound and of public health importance given the prevailing global health situation i.e. COVID-19 pandemic however, the authors did not identify the report as a systematic review in the title, did not include a systematic review registration number and did not include a completed checklist for reporting systematic reviews.
---

REVIEWER	Emiko A. Tajima University of Washington, USA
REVIEW RETURNED	07-Nov-2020

GENERAL COMMENTS	This study provides an informative overview of efforts to prevent or respond to violence against women and children in Europe in the context of the Covid-19 pandemic. The paper could be strengthened further by addressing the following:  1. It seems that the search terms were comprehensive for the review of publications, but less so for review of media. For the latter, I believe the search term was limited to "violence" -- please clarify this, and if this is correct, explain why only "violence" was used, and not "abuse" and other terms. In particular, I would imagine that searching only for "violence" would yield very little relating to child abuse. 2. The authors note that information was not fact checked. I am curious whether the authors have any reason to believe that any of the information was due to misinformation? 3. The majority of responses seem to represent public information campaigns and dissemination of helpline information. It would be helpful to offer more examples of the public service messages that were conveyed, including the range of the types of messages, and whether there were any conflicting messages. 4. It is notable that 42% of the initiatives / responses were policy changes. It would be interesting to offer more examples of the policies which were passed / enacted, and to know whether they were time limited (just during the pandemic) or whether they were permanent policy and practice reforms (such as the police detention of DV offenders for 48 hours instead of 3 hours, or mandatory reporting of VAWC). 5. Since this study is focused on violence against women and children, it would be helpful to offer more examples of efforts to respond to or prevent child abuse -- as it stands, the paper's findings are more centered on violence against women.
--

REVIEWER	Cathy Humphreys University of Melbourne
REVIEW RETURNED	08-Nov-2020

GENERAL COMMENTS	This is a relevant and timely article outlining a range of measures that countries across Europe and Russia have taken to address VAWC during COVID. There are a number of issues to address: Methodology: It would be good practice to reference theoretically the approach to the literature review e.g. a scoping review, a systematic review, a narrative review, a critical interpretive synthesis?? Situating where your approach is drawn from is helpful for readers. An example lies with the section on Quality Assurance. A reference to the literature re checking between authors is a practice that should be referenced. I was left wondering what actually constituted a media report? For example, I would have thought that some media reports would be reporting on research/reports that had been written and would
---

	therefore overlap with publication reports. Did this occur? I assume also that at times in the media reports that a lot of statistics drawn from government and NGO databases would be reported. Some of these would be publications. I would recommend greater clarity and possibly a mention of any overlap. Results: The results and the reporting through the Tables was descriptive and relevant. However, it was a bit 'all over the place' reporting at times by country, at other times by particular measures taken. I wondered whether it would be possible to organise under the headings of primary, secondary, tertiary measures using a public health approach. There are clearly overlaps between different levels but it might provide a clearer structure to this section. Possibly the same approach to structure could occur in the Discussion. The points about absences, particularly in relation to children are well made. I wonder if other issues of intersectionality were also lacking in terms of focusing on reaching specific populations. A few typos/grammar:  • The article opens using the acronym VAWC. When this changes to VAW and VAC then these terms need to be written in full initially. Alternatively stick with VAWC. • P9 line 8 – was the measure popular with teenagers generally or teenagers living with DV. Clarification would be helpful • P13 line31 – 'was' should be 'were' I think to qualify 'measures'
--	---

VERSION 1 – AUTHOR RESPONSE

Reviewer: 1

Comments to the Author

Thank you for asking me to review this paper. It reflects an interesting project that has been undertaken amazingly quickly. I have two more substantial comments. The first relates to the framing as VAWC. As far as I can tell from the text there were very few measures that specifically were designed to reach out to children and where these existed they seem to be for rather older children. There is some discussion of lack of work identifying abused children in school as these were closed and on parenting, however this seems rather uncomfortably placed in the paper which clearly doesn't refer much to measures for children at all. The question for me is: was very little done about VAC or was it missed in the search? Perhaps not so often reported in papers?

My perception is that VAC has been little discussed in the context of lockdown and these findings seem to support that. I think this is rather an important problem (for children) and that the authors should find a way of separating discussion of measures for VAW and VAC in the results so that the lack of VAC measures becomes more visible. Further just how much of a problem this will be for children (locked at home with abusers) should be separately discussed and recommendations framed.

When we conducted this study, we also realised that violence against children and especially preventive or response measures were rarely mentioned, or if they were mentioned, it was in the context of 'women and children facing abuse' and linked to broad measures to prevent or respond to 'domestic violence'. As you mentioned, we believe this is due to the fact that with schools closed, the main channel for reporting child abuse is compromised, and because child abuse prevention measures are often directed upstream such as parenting measures, which our search would not have found due to its focus on violence-related terms. Whilst conducting this review, we also realised this

focus to be on violence against women, which is a key finding, as we believe it is truly reflective of the situation and that the issue of violence against children during COVID-19 was, and is still, not widely considered.

In most reports, the type of violence was reported as 'domestic violence', which was mostly used without clarification as to whether it referred to violence against both women and children, or only to women. We did not analyse our data based on the number of responses from each country, so we are unable to provide a breakdown of the specific number of child-focused measures versus adult-focused.

Yet, to highlight the finding on the lack of responses to the perceived increase in VAC during COVID-19, and that those are mainly focused on teenagers, we have included an additional paragraph at the end of the results section. Plus, we have edited the part of the discussion that mentioned the lack of VAC to put more emphasis on the fact that children were so under-represented and that they must be considered going forward. We also amended a sentence in the conclusion to reflect this.

The second comment relates to VAW prevention. Most of the measures were ones that generally are considered 'response measures'. They may have enabled some individual women experiencing violence to get help, but other prevention programming this year has been largely halted across the world and none of these measures were directed at getting men to stop being abusive. Of great concern is that poverty is a potent driver of gender inequality and violence and this has increased this year. The main 'prevention' measure was awareness campaigns, which on their own have never been shown to prevent VAWC. I feel these concerns need to be built into the paper, as the suspension of work on prevention of VAWC, in the context of loss of gains on risk factors, will have a long term impact on the problem globally.

This is an important point and we have now dedicated a whole paragraph to it in the discussion section.

Table

Table 1 is first referred to in the methods and since it is conventional to place a Table after its first reference this is unsatisfactory. I suggest revising the section to remove reference to the table there so it can be clearly displayed in the results.

We have now deleted reference to Table 1 in the methods.

In Table 1: I am not clear how 'EU funding' is a measure— what was the EU funding for?

Changed to "EU funding earmarked for VAWC services"

Similarly I am not sure that community/corporate fundraising is a measure – again the question is – for what?

Changed to "Community/corporate fundraising for VAWC services"–

Minor edits

The paper has a number of typographical errors, a few have been identified below. It would benefit from a thorough edit.

The following errors highlighted by Reviewer 1 have now all been corrected in the manuscript:

Delete the second 'measures' in line 16 abstract

Article summary line 5 'a' before systematic

Line 13 – change 'meaning' to 'thus'

Page 5 line 3 'VAWC' is an important...issue.

Line 7 change 'or' to 'that is'

Line 9 'quarantine'

Line 23 delete 'to' after governments before enforce

Line 25 delete 'to' after governments before 'make'

Line 32 change 'to provide' to 'providing'
Page 6 line 2 change 'studies' to 'reports'
Page 8 line 1 'of' Kosovo
Page 8 sentence covering lines 6-11 needs editing
Line 17 – is it Justice and Equality?
Line 31 delete 'have'
Line 36 'were' rather than 'have been'
Line 38 delete 'who have being'
Page 9 line 2 'technology' rather than 'technical'
Line 7 is 'Telegram' correct here? – yes telegram is an App
Line 12 – is it numbers of relevant service providers?
Line 15 'using' rather than 'of'
Line 20 full stop after 'them' new sentence for the Andora example
Line 32 – delete 'the' and edit 'legal processes'
Line 40. Full stop after 'survivor' delete 'and' and capital for 'In'
Line 44 delete 'a'
Page 10 line 13 add 'the' before 'social media' – who was it who checked in? was it really 'the app'? if so how did that work
Page 12 line 12 insert 'often' before 'reported'
Line 23 'service provision'
Line 41 insert 'was' before 'most often'
Page 13 line 3-7 'As schools across the member states are now reopening, this will be a crucial time to ensure that children are offered the necessary support and services that they may not have had access to under the pandemic restrictions.' This sentence needs editing to take into account the fact the paper is being considered for publication in a different point in time

Reviewer: 2

Comments to the Author

The work contained herein is scientifically sound and of public health importance given the prevailing global health situation i.e. COVID-19 pandemic however, the authors did not identify the report as a systematic review in the title, did not include a systematic review registration number and did not include a completed checklist for reporting systematic reviews.

Thank you for this important clarification. This article is not based on a systematic literature review in the traditional sense but combines a systematic newspaper search and a scoping review of available literature on an emerging subject. To avoid confusion, we have removed the term systematic from the methods section and now refer to the review as a scoping review.

Reviewer: 3

1. It seems that the search terms were comprehensive for the review of publications, but less so for review of media. For the latter, I believe the search term was limited to "violence" -- please clarify this, and if this is correct, explain why only "violence" was used, and not "abuse" and other terms. In particular, I would imagine that searching only for "violence" would yield very little relating to child abuse.

The search terms for the media review were as comprehensive as for the publication review and can be found in the Supplementary Methods. They were: Coronavirus and abuse: (Covid OR Covid19 OR Coronavirus OR corona OR SARS-CoV-2 OR Covid 19) AND ("domestic violence" OR "domestic abuse" OR "intimate partner violence" OR "Gender based violence" OR "sexual violence" OR

"femicide" OR "child abuse" OR "child maltreatment" OR "child neglect" or "child exploitation" OR "bullying" OR "trafficking" OR "sexual exploitation" OR "sexual abuse" OR "child marriage" or "youth violence" OR "infanticide" OR "stalking")

2. The authors note that information was not fact checked. I am curious whether the authors have any reason to believe that any of the information was due to misinformation?

This statement was included as we did not verify whether the measures reported in the newspaper searchers were actually implemented and sustained in the way they were reported. For example, while Israel reported to provide domestic violence survivors with a hotel room, our key informants told us in an interview (not part of this paper) later that this measure has been overturned shortly afterwards to give the hotel rooms designated to domestic violence survivors to people who needed to isolate due to COVID-19.

It is beyond the scope of this paper to verify all newspaper stories. Originally, we had used the term 'reportedly' before each example measure was given, for example, 'the government of France reportedly provided hotels for domestic abuse survivors', but this impacted the flow of the paper and considerably increased the word count. We therefore included the fact checking statement instead. The importance of the statement was stressed by our collaborators at the WHO who wanted to ensure that we were not reporting any measures as definite facts. As most of the media reports we read included quotes from politicians and service providers, I do not have reason to believe that any of the information was based on misinformation. We have, however, now amended the sentence slightly to not imply that we do not believe them, as 'fact-checking' can be a contentious word these days with reports of 'fake news' etc and this is a debate we do not want to enter here. Thank you for bringing this to our attention.

3. The majority of responses seem to represent public information campaigns and dissemination of helpline information. It would be helpful to offer more examples of the public service messages that were conveyed, including the range of the types of messages, and whether there were any conflicting messages.

As we were studying reports of measures used, often the newspaper reports would simply include the name of the campaign and not provide further details about it. To analyse the content of such campaigns would certainly be a very interesting study, however, was beyond the scope of this paper.

4. It is notable that 42% of the initiatives / responses were policy changes. It would be interesting to offer more examples of the policies which were passed / enacted, and to know whether they were time limited (just during the pandemic) or whether they were permanent policy and practice reforms (such as the police detention of DV offenders for 48 hours instead of 3 hours, or mandatory reporting of VAWC).

This is a very good point, and we agree that more details on policies would be interesting. However, we were very limited by the word count with this paper, so could only provide a few examples of each measure and we did not want to give more weight to government-led measures at the expense of civil society-led measures. With regards to the longevity of new policies, this kind of detail was generally not provided in the media reports and would involve further research into each policy, which was beyond the scope of this paper. However, this research is part of a wider research project that involved key informant interviews, and these provided further details and critical analyses of such measures. This is still in process but once published will provide a more detailed examination of this. We will take your important point into account for this analysis.

5. Since this study is focused on violence against women and children, it would be helpful to offer more examples of efforts to respond to or prevent child abuse -- as it stands, the paper's findings are more centered on violence against women.

Please also see our response to Reviewer 1's similar important comments on this and our changes implemented in response to those. This, unfortunately, was the case with our findings. As stated in the paper, the large majority of media reports simply used the term 'domestic violence' to refer to violence against women and or children. Where examples that related to children specifically were found, we made the effort to ensure these were included in the paper but as you can see from the results, this was not common. We have discussed the possible reasons for this in the discussion, particularly, the fact that many child abuse prevention measures are not explicitly linked to abuse, and hence would not have been found using our search strategy. However, we do believe this accurately represents what was present by the media during this time, as we found that media reports were greatly focused on violence against women, rather than children.

Reviewer: 4

Comments to the Author

This is a relevant and timely article outlining a range of measures that countries across Europe and Russia have taken to address VAWC during COVID. There are a number of issues to address:

Methodology: It would be good practice to reference theoretically the approach to the literature review e.g. a scoping review, a systematic review, a narrative review, a critical interpretive synthesis?? Situating where your approach is drawn from is helpful for readers.

Please see our response to reviewer 2. We have now updated to clarify that this was a scoping review.

An example lies with the section on Quality Assurance. A reference to the literature re checking between authors is a practice that should be referenced.

I was left wondering what actually constituted a media report? For example, I would have thought that some media reports would be reporting on research/reports that had been written and would therefore overlap with publication reports. Did this occur? I assume also that at times in the media reports that a lot of statistics drawn from government and NGO databases would be reported. Some of these would be publications. I would recommend greater clarity and possibly a mention of any overlap.

This is a good point. We did not notice many media searches that quoted publications, in fact, it was the publications that referenced back to the news reports. Only seven publications were included in this study and none included primary data collection, nor did any contain analyses of measures taken to prevent and respond to VAWC during COVID-19, hence most were also reporting on media reports and statements posted online. For example, the study may look at service demand during the pandemic, and their introduction may include some examples of measures announced by governments, we would extract these measures to add to our report if we did not already have them from the media search. Most often, the media reports were interviews with service providers and quotes from government officials, which provided the bulk of our results, the publications just supplemented a few additional measures that our search had not identified.

We agree it is important to make this clear in the results, so we have added the following statement: "The publications found did not include any primary data collection regarding measures to prevent and respond to VAWC during COVID-19. They did, however, cite additional measures announced through media platforms, most often online newspaper articles. Due to this, the publications were

used to supplement the results of our media search, with additional media reports cited through the publications, and did not offer any examples of primary data analyses.”

Results: The results and the reporting through the Tables was descriptive and relevant. However, it was a bit ‘all over the place’ reporting at times by country, at other times by particular measures taken. I wondered whether it would be possible to organise under the headings of primary, secondary, tertiary measures using a public health approach. There are clearly overlaps between different levels but it might provide a clearer structure to this section. Possibly the same approach to structure could occur in the Discussion.

We have decided to structure the results under the headings of Government-led/sponsored and NGO or civil society-led/sponsored to highlight what governments were doing versus what civil society were doing, as this would allow us to call for further policy changes and investment from governments as a direct policy output, plus to highlight the effect some government policies have on the service provision of civil society organization. We also had the aim to highlight what types of measures were being used, and by whom they were led (gov versus non-gov) to encourage inspiration for the second or third waves of COVID-19 and future pandemics. This is why table 1 is not broken down by country, and instead is comparing government and non-government led measures. If we were not separating by government/non-government then I agree that by the level of measure, as you suggested, would be the most pragmatic way to structure the results. However, if we were to implement this structure under the two headings of governmental and non-governmental, I believe the paper would become very repetitive, as it also would having a government/non-governmental breakdown within each measure level.

The points about absences, particularly in relation to children are well made. I wonder if other issues of intersectionality were also lacking in terms of focusing on reaching specific populations.

We barely identified measures targeting those groups particularly at risk of VAWC during COVID-19 or in general and have highlighted this in an additional sentence in the end of the discussion. It is an important factor to highlight, thank you.

A few typos/grammar:

- The article opens using the acronym VAWC. When this changes to VAW and VAC then these terms need to be written in full initially. Alternatively stick with VAWC

This has now been changed

- P9 line 8 – was the measure popular with teenagers generally or teenagers living with DV.

Clarification would be helpful

This is a good example of where we were limited by the level of detail in the media reports. The news report simply states “The chatbot has been popular among teen users.” When we encountered information like this, we tried to relay exactly what was presented in the media report, to avoid creating our own narrative of the situation. To help clarify this, I have changed it to the following: “this measure was reportedly popular amongst teen users”, as I think this conveys what the report stated better. Thank you for highlighting the ambiguity here.

- P13 line31 – ‘was’ should be ‘were’ I think to qualify ‘measures’

This has now been changed

Thank you very much for all of your very helpful feedback.

Best wishes,

Isabelle Pearson, Nadia Butler, Zhamin Yelgezekova, Åsa Nihlen, Isabel Yordi Aguirre, Zara Quigg and Heidi Stöckl,

VERSION 2 – REVIEW

REVIEWER	Rachel Jewkes South African Medical Research Council
REVIEW RETURNED	26-Nov-2020

GENERAL COMMENTS	I am very happy with the way the authors have addressed the reviewers' comments
---

REVIEWER	Emiko Tajima University of Washington USA
REVIEW RETURNED	10-Dec-2020

GENERAL COMMENTS	No additional comments. The authors have addressed my concerns.
---

REVIEWER	Cathy Humphreys University of Melbourne, Australia
REVIEW RETURNED	06-Dec-2020

GENERAL COMMENTS	I believe that the authors have addressed the reviewer comments carefully. The concerns that I raised have all been addressed, and there is a well considered response to the other reviewers. This will be a helpful article providing foundational material for other researchers and practitioners
---